# “A Huge Gap”: Health Care Provider Perspectives on Cancer Screening for Aboriginal and Torres Strait Islander People in the Northern Territory

**DOI:** 10.3390/ijerph21020141

**Published:** 2024-01-27

**Authors:** Emma V. Taylor, Sarah Dugdale, Christine M. Connors, Gail Garvey, Sandra C. Thompson

**Affiliations:** 1Western Australian Centre for Rural Health, The University of Western Australia, Geraldton, WA 6530, Australia; sandra.thompson@uwa.edu.au; 2Health Statistics and Informatics, NT Health, Darwin, NT 0800, Australia; sarah.dugdale@nt.gov.au; 3Public Health Division, NT Health, Darwin, NT 0800, Australia; christine.connors@nt.gov.au; 4The School of Public Health, Faculty of Medicine, The University of Queensland, Herston, QLD 4006, Australia; g.garvey@uq.edu.au

**Keywords:** cancer screening, primary health care, Indigenous, Aboriginal and Torres Strait Islander, remote, Australia, bowel cancer screening, colorectal cancer screening, breast cancer screening, cervical cancer screening

## Abstract

Cancer is one of the leading causes of death for Aboriginal and Torres Strait Islander people in the Northern Territory (NT). Accessible and culturally appropriate cancer screening programs are a vital component in reducing the burden of cancer. Primary health care plays a pivotal role in facilitating the uptake of cancer screening in the NT, due to the significant challenges caused by large distances, limited resources, and cultural differences. This paper analyses health care provider perspectives and approaches to the provision of cancer screening to Aboriginal people in the NT that were collected as part of a larger study. Semi-structured interviews were conducted with 50 staff from 15 health services, including 8 regional, remote, and very remote primary health care (PHC) clinics, 3 hospitals, a cancer centre, and 3 cancer support services. Transcripts were thematically analysed. Cancer screening by remote and very remote PHC clinics in the NT is variable, with some staff seeing cancer screening as a “huge gap”, while others see it as lower priority compared to other conditions due to a lack of resourcing and the overwhelming burden of acute and chronic disease. Conversely, some clinics see screening as an area where they are performing well, with systematic screening, targeted programs, and high screening rates. There was a large variation in perceptions of the breast screening and cervical screening programs. However, participants universally reported that the bowel screening kit was complicated and not culturally appropriate for their Aboriginal patients, which led to low uptake. System-level improvements are required, including increased funding and resourcing for screening programs, and for PHC clinics in the NT. Being appropriately resourced would assist PHC clinics to incorporate a greater emphasis on cancer screening into adult health checks and would support PHCs to work with local communities to co-design targeted cancer screening programs and culturally relevant education activities. Addressing these issues are vital for NT PHC clinics to address the existing cancer screening gaps and achieving the Australian Government pledge to be the first nation in the world to eliminate cervical cancer as a public health problem by 2035. The implementation of the National Lung Cancer Screening Program in 2025 also presents an opportunity to deliver greater benefits to Aboriginal communities and reduce the cancer burden.

## 1. Introduction

Cancer is one of the leading causes of death for Aboriginal and Torres Strait Islander people in the Northern Territory [1,2] (hereafter the term ’Aboriginal’ is respectfully used). Like the broader trend in Australia, the gap in cancer mortality rates between Aboriginal and non-Aboriginal populations in the NT is widening [3]. While the cancer mortality rate in the non-Aboriginal population has steadily improved, Aboriginal peoples in the NT have experienced a yearly increase of 1% in cancer mortality from 1991 to 2015 [3]. Aboriginal people in the NT have a higher incidence of several high fatality cancers, including liver, lung, and other smoking-related cancers, than non-Aboriginal people, and the incidence of breast, bowel, and prostate cancers is increasing [3,4]. Screening (both population-based and targeted risk-based) remains an important means of detecting cancer early, at a point that can result in a cure or markedly impact any progression of disease [5,6]. However, cancer screening participation is generally lower among Aboriginal peoples in the NT, and Aboriginal Territorians are more likely to be diagnosed at a more advanced stage [7]. Early detection is key to improving outcomes necessitating attention on improved screening services for this group [8].

Australia has national population-based cancer screening programs for cervical, breast, and bowel cancer [9]. The National Cervical Screening Program (NCSP) began in 1991; the BreastScreen Australia program was introduced between 1991 and 1995; and the National Bowel Cancer Screening Program (NBCSP) started in 2006 but was not fully implemented until 2020 [10].

When the NCSP began, it was using the two-yearly Papanicolaou (Pap) smear, or ‘Pap test’. However, in December 2017 it was replaced with the human papillomavirus (HPV)-based Cervical Screening Test (CST), the screening interval was increased to five years, the eligible age shifted to 25–74 years and the option of self-collection was introduced (although initially only for under-screened or never-screened women) [11,12]. In the NT, cervical screening is available from primary health care services including Aboriginal Community Controlled Health Services (ACCHS), GP clinics, and health centres, and while the test is free for eligible people with a cervix, the health practitioner may charge a fee for the appointment [13]. Cervical cancer incidence in the NT has decreased by more than 50% between 1991–1995 and 2011–2015, indicating the effectiveness of the cervical screening program, and participation in cervical screening in the NT is comparable to the national average (61.7% compared to 62.4%) [9,14]. However, the incidence of cervical cancer remains 60% higher for Aboriginal women compared to non-Aboriginal women in the NT [9].

BreastScreen Australia is the national breast cancer screening program, which actively offers women aged 50–74 years free mammograms every two years, as well as providing free mammograms to women aged 40–49 and over 74 years. In the NT, access to breast screening varies, with permanent facilities in the outer regional areas Darwin and Palmerston. Screening is available during two six-week periods a year in Alice Springs, and a four-wheel drive bus provides screening to fifteen remote Aboriginal communities across the NT, as well as the large remote towns of Tennant Creek, Katherine and Nhulunbuy (Gove) [15]. Breast cancer is the most common cancer for females in the NT, and breast cancer screening participation in the NT is the lowest in Australia (36.4% compared to 49.9%) and is lower again for Aboriginal women (26.7%) [9,14].

The NBCSP invites eligible Australians to screen for bowel cancer using a free home immunochemical faecal occult blood test (iFOBT) kit which is mailed to their residential address every two years. When the NBCSP began, eligible participants in two age bands, 55 and 65 years, were offered a one-off test, and in 2008, a one-off test for people turning 50 was added to the program; from 2013, the program was slowly expanded with biennial screening of all Australians aged 50–74 by 2020 [16]. Another recent change to the program allows health care providers to bulk order NBCSP kits and issue them directly to eligible patients. However, because the at-home bowel screening kits are temperature sensitive, distribution of kits in the NT is limited by the hot-zone policy [17]. This policy restricts the distribution of kits when an area’s average monthly temperature is above 30 degrees Celsius and limits kit distribution in some NT postcodes to only three months per year [17]. Despite bowel cancer being one of the most prevalent cancers for the NT non-Aboriginal population and increasing over the last thirty years among the NT’s Aboriginal population, bowel cancer screening participation in the NT is the lowest in Australia (25.3% compared to 40.9%), and is even lower for Aboriginal people [9,14]. It should be acknowledged that the NBCSP was launched 15 years after the other national population-based screening programs and had an incremental roll out. This delayed comprehensive education campaigns directed at the public and at health professionals. It is also the only self-administered population screening test and the first population screening test to include men, which may have contributed to the lower participation rate [10].

The three cancers for which population screening occurs in Australia—bowel, breast, and cervical cancer—represent one-third of the incident cancers in the NT in 2018 [2], and the effective implementation of these screening programs in the NT is an important component of improving cancer outcomes. Furthermore, the screen-detectable cancers—lung, liver, and prostate cancer—are the first, second, and fifth most common cancers among NT Aboriginal men, and lung cancer is the second most common cancer for NT Aboriginal women [2]. Primary health care plays a pivotal role in facilitating the uptake of population-based screening and in implementing targeted risk screening in Australia [18,19,20]. There are unique challenges in delivering effective cancer screening programs in the NT: the population is geographically dispersed over a vast area, with parts of the territory inaccessible at certain times of year due to extreme weather conditions; large distances between cancer screening locations; a culturally and linguistically diverse Aboriginal population, 75% of whom live in remote or very remote areas; and a health workforce with high turnover [21,22]. Health system factors, such as the availability of health care providers and access to screening services and health practitioner attitudes and knowledge of cancer and screening programs may also impact screening [23,24]. In addition, several factors affect Aboriginal people’s willingness to present and participate in screening, including feelings of shame, fear of diagnosis and distrust of the health service, lower health literacy, logistical difficulties accessing screening services, and competing priorities [25,26]. Enablers of screening have been found to include culturally safe services, flexible service delivery, the availability of Aboriginal health staff, and positive relationships with health services and staff [25,27]; therefore, health care services and staff have an important role to play in improving screening rates.

There is a scarcity of research that has explored the perspectives of health care providers on the delivery of screening to Aboriginal peoples, particularly in the NT. Health care provider perspectives may offer insights into screening provision and health practitioner attitudes, as well as identifying ways to increase screening participation. This study aimed to analyse health care provider perspectives and approaches to the provision of screening to Aboriginal Australians in the NT.

## 2. Methods

### 2.1. Ethical and Cultural Considerations

Approval to conduct this study was granted by the Human Research Ethics Committee at the University of Western Australia (RA/4/1/6286), the Central Australian Human Ethics Committee (HREC-13-198) and the Human Research Ethics Committee of the Northern Territory Department of Health and Menzies School of Health Research (2016-2576).

This study forms part of a national project to identify and describe cancer services providing treatment to Aboriginal cancer patients in Australia. It was a component of research within the Centre for Research Excellence (CRE), Discovering Indigenous Strategies to improve Cancer Outcomes Via Engagement, Research Translation and Training (DISCOVER-TT) funded by the National Health and Medical Research Council (NHMRC). The CRE was led by an Aboriginal researcher and brought together Aboriginal and non-Aboriginal researchers from around Australia, with the aim of improving outcomes and services for Aboriginal people with cancer. We adhered to the NHMRC Guidelines for Ethical Conduct in Aboriginal and Torres Strait Islander Health Research [28]. An Aboriginal Advisory/Reference Group was formed prior to commencing this study which provided advice and support to the study. In addition, we consulted with local Aboriginal Community Controlled Health Services (ACCHS).

The research team involved in the analysis consisted of five women (EVT, SD, CMC, GG and SCT). GG is a Kamilaroi woman and professor of Indigenous Health Research, with extensive research expertise in improving outcomes for Aboriginal people with cancer. Three of the non-Indigenous team members have clinical backgrounds (SD, CMC and SCT). SD and CMC are based in the NT and have extensive experience in the provision of Aboriginal health care, primary health care, and expertise in cancer screening. SCT and EVT have combined experience with collaborative research into improving Indigenous health outcomes of over thirty years.

### 2.2. Participants and Recruitment

Health professionals and support staff (Aboriginal or non-Aboriginal) in the NT were eligible if they were involved with the treatment, care, or support of Aboriginal cancer patients, if they filled a leadership role in the care of Aboriginal cancer patients, or if they provided primary health care to Aboriginal patients and their families. Participants were purposively recruited using a snowballing strategy to achieve maximum variation sampling with the most divergent forms of experience. Potential staff participants were approached in clinical service areas and at staff meetings and were invited to participate through personal invitation. Study information was provided to potential participants in-person or via email. Some potential participants forwarded the study information to colleagues or suggested additional potential participants to the investigators. As data collection progressed and gaps in participant groups became evident, efforts became more focused on capturing the voices and experiences of participants that were not well represented in the data. Participation was voluntary and written or oral consent was provided by all the participants prior to the interview.

### 2.3. Data Collection

Semi-structured interviews were conducted between September 2016 and April 2019, by one researcher, with the majority conducted in-person at the participant’s workplace. A small number were conducted via telephone due to issues of availability and geographic distance. In one instance, two participants requested a joint interview. The interviews ranged from 20 min to 2 h 45 min, and generally lasted just over an hour. Some interviews in the workplace were interrupted due to work demands and continued later. All the interviews were audio-recorded, transcribed verbatim by two team members, and checked against the recordings for accuracy, except for one interview where an equipment failure was detected after the interview concluded. In that case, the extensive notes taken during interview were used.

The interviews were guided by a set of broad, open-ended enquiries including questions on the staff member’s role in providing care to Aboriginal people, the typical pathway for Aboriginal cancer patients, how decisions about cancer treatment are made, links between primary and tertiary health services, and questions around primary prevention, early detection and screening, community education and end-of-life- care. The interview guide for health professionals is presented in Appendix A. Information on age, gender, Indigenous status, description of role, length of time working at the health service and years of experience providing care to Aboriginal patients was collected during the interview. The geographic remoteness of the NT health services and participants were categorised as outer regional (Darwin and surrounds), remote, or very remote using the Australian Statistical Geography Standard (ASGS) Edition 3 [29].

### 2.4. Data Analysis

We followed the thematic analysis process described by Green et al. [30] of immersion in the data with rereading, coding (using a hybrid inductive–deductive coding strategy), categorisation, and aggregation of identified themes. Transcripts were de-identified and imported into NVivo 13 for initial data organisation. Transcripts were reviewed by three team members. Analysis was iteratively conducted through line-by-line review of the transcripts by one researcher (EVT) to identify themes. Discussions and additional analysis between SCT and EVT refined themes and triangulated staff interviews.

The data were then analysed and grouped according to the seven steps of the *Optimal Care Pathway for Aboriginal and Torres Strait Islander People with Cancer* [20] by EVT. This Cancer Australia publication was chosen as a tool for analysis because it provides guidance to health services and health professionals on best practice care of Aboriginal people with cancer throughout the patient journey [20]. The seven steps represent seven critical stages in the patient journey, starting with prevention and early detection (which includes screening) and continuing through to diagnosis, treatment and end-of-life care. During this analysis process, screening was identified as an area of interest due to the rich data collected, despite it not having been a focus initially. Key interviews were then re-read and manually coded by EVT to develop existing themes and identify additional themes with a focus on screening. Further refinement of themes to reach agreement on the final themes occurred within the team. Analysis was documented to ensure that each theme could be traced back to the original data, using direct references from the interviews to provide evidence and maintain the voice of the participants. Data interpretation and recommendations arising from the data were checked with key stakeholders involved with cancer screening in the NT and cancer screening in the Aboriginal population and additional information incorporated into the final analysis and recommendations.

## 3. Results

### 3.1. Participant Characteristics

Fifty participants from 15 health services were interviewed. Health services consisted of primary health care (PHC) clinics (n = 8), hospitals (n = 3), a cancer centre (n = 1), and support services (n = 3) (Table 1). Most of the services (n = 10, 66%) were located in remote or very remote areas, with the remaining third (n = 5) located in the outer regional area of Darwin and surrounds. All PHC clinics were operated by NT Health.

Almost half the participants (n = 23, 46%) lived and worked in a very remote location, with 13 (26%) living in a remote area, and 14 (28%) living in the outer regional area of Darwin and surrounds. Participants included more women (n = 38, 76%) than men, 33 (66%) were aged between 40 and 59 years of age, and 14 (28%) identified as Aboriginal or Torres Strait Islander. Over half (n = 27, 54%) had been in their current position less than 5 years (range 3 months to 33 years). Participants had on average 10 years of experience providing care to Aboriginal patients (range 6 months to 33 years). Two-thirds (n = 33, 66%) of participants worked in primary health care. A diverse set of professions participated, including Aboriginal Health Practitioners (AHPs), Aboriginal Liaison Officers (ALOs), registered nurses (RNs), general practitioners (GPs), physicians, social workers, dietitians, managers, and administrative staff. Some participants held dual roles (e.g., cancer care coordinator and palliative care nurse).

### 3.2. Health System Factors: “A Huge Gap”

Some staff (n = 5) perceived cancer screening as a “huge gap” in the NT health system. One remote hospital-based staff member felt that screening was particularly under resourced for the Aboriginal population which led to poorer outcomes: “*I think if we were to put in as much effort [into screening] as we do with our non-Indigenous clients, then our Indigenous clients, I think we would see a different trend*” (Remote Hospital Cancer Care Coordinator HP049).

Staff observed that funding, programs, and support were more readily available once a cancer diagnosis was made, particularly for certain cancers such as breast cancer, but accessing screening was challenging for people living in remote communities. Several participants highlighted the financial burden to patients of accessing screening, as travel for screening tests was not funded by the NT’s Patient Assisted Travel Scheme (PATS). This resulted in people not getting screened because they could not afford to or did not have access to transport to travel to the nearest town. As one RN in a very remote community explained:

*“There’s no funding for patient travel money to get to screening, so you have to fund yourself… So here we know this population are late diagnosis probably in most cancers but to get early diagnosis is a big gap. That’s where it is, it’s not funded and yet there is this wave of interventions and things that happen post diagnosis but the stuff at the front is difficult”* (Very Remote PHC RN HP032).

One RN at a regional hospital gave the example of the high incidence of Hepatitis B virus (HBV) in the NT, which increases the risk of hepatocellular carcinoma (HCC). National guidelines recommend screening all people with chronic HBV who are Aboriginal and over the age of 50 years every 6 months; this requires an ultrasound and a blood test [31,32]. However, the RN stated that it’s “*not resourced in the NT at all*” as there were very few ultrasonographers working in the remote communities and very few of the PHC clinics had access to an ultrasound machine. Therefore, the guidelines were not able to be met “*maybe [eligible patients] have an ultrasound scan maybe once a year at best, maybe they don’t even get one… I don’t think the screening is done enough and systematically*” (Regional Hospital RN HP011).

### 3.3. Health Service Factors: “We Are Probably the Only Clinic That’s Doing It”

Multiple staff (n = 8) at two PHC clinics believed that screening was an area in which they were doing well, with systematic screening, targeted programs, education, and high screening rates. Staff at both clinics described routinely and “automatically” screening for different cancers as part of all adult health checks. In one remote clinic, the routine screening was instigated by the arrival of a GP who had incorporated a greater focus on screening into their consults and expected other staff at the clinic to do the same. Staff reported that it had become standard practice to include a bowel screen and, for men, a prostate cancer screen, in all over 50 adult health checks. However, this was perceived as unusual, with the manager of the clinic observing “*most places that’s not being done, we are probably the only clinic that’s doing it*” (Remote PHC Manager HP025).

In addition to incorporating routine screening as part of health checks, staff at a very remote clinic described having a big focus on promoting screening programs through education and community visits. One staff member ascribed their high breast screen and cervical screening rates to a focus on education and community engagement. “*Women’s health is pretty big [here]. There’s a lot of talk and education around women’s health*” (Very Remote PHC Admin HP041). A GP registrar at the same clinic described how male PHC staff travelled to a different remote community each month to conduct “men’s health” clinics, including health checks, chronic disease management, and education.

### 3.4. Health Care Provider Attitudes: “Not on the Highest of Priorities”

GPs and nurses (n = 4) at three PHC clinics did not see cancer screening as a priority due to the burden of acute and chronic disease, lack of resourcing and their perception of the low incidence of cancer in the Aboriginal community. High staff workload negatively impacted capacity to screen. As one GP in a very remote community explained, “*[we are in] survival mode every day in clinic, just trying to treat really sick people—get through this box of things, I mean no-one even opened the box of bowel screen stuff in the office*” (Very Remote PHC GP HP043). This lower prioritisation of cancer screening was exacerbated in clinics with a high turnover of staff, where staff reported on the demands of acute presentations (broken arms, infections) with consequently less attention on chronic diseases and wellness checks (which often incorporated screening). High turnover, exacerbated by subsequent reliance on short-term, agency or locum staff, was felt to negatively impact patients’ trust in clinic staff and their willingness to present for screening. Of those interviewed, 12% (n = 6) had been in their current position for less than 1 year and 22% of participants working in very remote locations had been in their current position for less than 1 year. One staff member suggested training AHPs to provide cancer screening, believing they would provide a more stable workforce and have a greater vested interest in improving cancer outcomes in their communities.

Several staff believed that there was not a high incidence of cancer in their community compared to other conditions, and this impacted the priority they placed on screening. A GP in a very remote community explained that, due to the “*enormous burden of chronic disease, and cancer because of its relative infrequency is not on the highest of priorities… until it kind of comes to a head. Like when we are palliating.*” (Very Remote PHC GP HP027). One RN in a remote community believed that more education was needed for health professionals because cancer screening was “*not really on people’s radar*” (Remote PHC RN HP014). This matched perceptions of screening by hospital staff, who described screening programs in PHC clinics as “*a bit haphazard*” (Remote Hospital Cancer Care Coordinator HP049) and not meeting guidelines, which they attributed to inadequate resourcing, high staff turnover, and burnout.

A small number of staff reported screening patients opportunistically during annual adult health checks. Two participants mentioned checking for weight loss, another mentioned checking for anaemia and asking patients whether they were up to date on their bowel, breast, and cervical screening. One RN commented that the health checks “*don’t really push you towards [cancer] screening*” (Remote PHC RN HP014) while an RN at a very remote clinic stated that opportunistically screening patients who presented for an unrelated reason felt uncomfortable and exhausting for the clinician and patients.

*“It’s just all so forced. It’s like, okay, I guess you’re here for this. And it’s just exhausting. So they might come in for a sore big toe and they end up with their adult health check, bloods and urine done. And a foot check and uh, we’ll do your waist randomly right now, <laugh>, you know it just doesn’t flow. Like it’s so disjointed”* (Very Remote PHC RN HP034).

Staff at several PHC clinics found it difficult to engage patients with screening, with one Indigenous GP registrar stating that “*people [in remote communities] are not that interested in screening*” (Very Remote PHC Indigenous GP Registrar HP035). Staff attributed their difficulties engaging patients with cancer screening to a belief that community members were not interested in “*worrying about the future*”. Many staff felt that poor health literacy was a barrier to community members understanding the importance of screening, especially when English was not a first language, and more community education was needed to address this.

There was minimal data about screening from regional hospital and cancer centre staff, apart from comments by two staff that Aboriginal people were under-screened due to lack of access to screening and reluctance to attend screening.

### 3.5. Bowel Screening: “A Shame Thing”

There was a strong opinion amongst the rural and remote PHC staff that the national bowel screening home test kit was inappropriate for their Aboriginal patients (n = 12). Multiple staff stated that Aboriginal people in the community viewed the test as “dirty” and “a shame thing” and would not feel comfortable with collecting a sample at home. Posting the kit was problematic as many members of the community did not have a fixed address or access to a mailbox, while others did not check their mail regularly. Staff commented that the test was too complicated, and the instructions that came with the test were a barrier, as English was often the second, third or fourth language spoken by their patients. Others reported that the test was frequently misunderstood, and that some community members thought it meant they had been diagnosed with cancer. Many staff reported that the test was thrown away because people were not sure what it was. As one Indigenous GP in a very remote community explained:

*“English is their second language… you get this wordy thing, you may or may not [be able to] read… you get this weird thing in the mail and you go ‘oh I don’t want to do that’, so a lot of people will just throw them out”* (Very Remote PHC Indigenous GP HP038).

Another GP in a very remote community felt frustrated when they contacted the bowel screening office on behalf of their patient and was told to refer their patient to the website for help. As the GP pointed out, many of their patients did not have a house, a phone, or access to the internet.

Some participants felt the test would be better managed if it was sent to the PHC clinic, for the clinic to follow up, explain and organise. The manager of one remote clinic described how one of their GPs, on noticing their patients were not comfortable completing the postal screening kit, had instigated a process whereby the clinic routinely included bowel screening as part of all adult health checks for those aged over 50 years. Several staff at the clinic reported that this approach appeared to be more acceptable to their patients than the postal kit, and more efficient in terms of following up results. The manager reported that the clinic had successfully screened a lot of the older men in their community through this process.

However, other participants said they did not know how to explain bowel screening to their patients and did not have the time or adequate resources to dispense bowel screening tests to their patients. Three GPs in two different communities felt unengaged with the bowel screening process and did not feel it was their responsibility, one describing bowel screening as “*not a branch of the clinic really. It’s just the government sends the [bowel screening] kit and what happens to the kits we really don’t know*” (Remote PHC GP HP013) and another commenting “*I’ve never seen bowel screening*” (Very Remote PHC Indigenous GP Registrar HP035). However, one GP was planning to ask the new chronic disease nurse who had just joined the clinic to target bowel screening “*because bowel screening is something that we are way behind*” (Very Remote PHC GP HP043). One GP Registrar and two nurses questioned whether bowel cancer incidence was high enough in the Aboriginal community to justify investing in bowel cancer screening.

### 3.6. Breast Screening: “The Breast Screen Bus, It Does a Fabulous Job”

Staff in the study had mixed perceptions of the BreastScreenNT bus. Several staff, including two Aboriginal staff, spoke positively of the bus, describing how it raised awareness and resulted in earlier diagnosis of breast cancer by going to communities every two years. “*The breast screen bus, it does a fabulous job*” (Remote Hospital Cancer Care Coordinator HP049). One RN in a very remote community planned promotions around the arrival of the bus to encourage women to attend, such as “*a BBQ and a big prize for the person who turns up first*” (Very Remote PHC RN HP029). However, several staff observed that the bus had specific power and water requirements, which limited where it could be located. In communities where the available infrastructure meant that the bus was parked in a highly visible location, this was considered to impact some women’s willingness to attend the bus due to feelings of shame. Staff commented that there should be more community consultation about appropriate locations for the BreastScreenNT bus. “*Actually go out to the communities, not just park up a bus and say, right, this is happening. But engage the communities to try and be invited in.*” (Remote Hospital Cancer Care Coordinator HP049) Health staff at one remote community had campaigned unsuccessfully for five years to get the bus to visit their community but they were not considered a priority because the nearest remote hospital was 90 km away: “*[the BreastScreenNT bus] will not pull up here, because our people can get themselves to [nearest town] … well they can’t*” (Remote PHC Manager HP025).

### 3.7. Cervical Screening: “We Talk to Them and Explain to Them the Reasons Why”

Staff including Aboriginal staff at three remote and very remote PHC clinics spoke positively about cervical screening and perceived it was an area they did well. Multiple staff described successful cervical screening programs run by the nursing staff, with one clinic proactively contacting eligible patients when a midwife visited the community and offering transport to the clinic. The AHP at this clinic responsible for contacting the patients would visit eligible women and let them know when it was time for their “women’s check” (as opposed to “adult health check”) and reported that patients felt comfortable with this approach. “*There was no shame job around it… they knew what they were coming in for. I would say it’s for your women’s check and they were comfortable with that. Almost all of them came in for their checks*” (Remote PHC Indigenous AHP HP019). Staff at this clinic reported that some patients preferred the screening to be conducted by a visiting or agency nurse because they knew the local nurses “*a little bit too well*” (Remote PHC RN HP022).

Staff at all three clinics emphasised the importance of education to improve cervical screening rates, including explaining to patients why screening was important. “*We talk to them and explain to them the reasons why*” (Very Remote PHC4 Indigenous AHP HP031). “*Because obviously if you understand why you should get it [cervical screening] done, it’s more likely you will go in and do it*” (Remote PHC RN HP022). An RN in a very remote community described how they demystified the procedure for patients “*When someone comes in here for cervical screening, I show them a speculum and I show them a cytobrush that I want to use*” (Very Remote PHC RN HP032). The RN emphasised the importance of ensuring patients understood what was meant by “screening”, as the term is frequently used when talking about STI screening, which can cause confusion with cancer screening:

*“I explain that I am wanting to look for something different to an STI and try and make sure that they actually do know why I want to do this… get mixed up with STIs and cervical screening. I’ve had young girls come in and say they want a women’s check, and when you actually talk to them its STIs they understood, not cervical screening”* (Very Remote PHC RN HP032)

Two of the clinics reported having only a small number of under-screened women. A GP registrar at a very remote clinic had conducted an audit to see how many of their patients would be eligible for self-screening under the recently introduced CST and found that only “*four or five*” had not had a Pap smear for over 4 years (Very Remote PHC Indigenous GP HP038).

This contrasted with GPs at a different very remote clinic, who felt that their patients were “*under screened*” (Very Remote PHC GP HP043) and observed that “*cervical screening doesn’t happen that often*” and women were screened “*a lot less*” than other places they had worked (Very Remote PHC Indigenous GP Registrar HP035). It was speculated that patients might be being screened at the nearest town (200 km away), and attributed patients’ unwillingness to privacy concerns and feelings of shame at having to undress.

Staff at all clinics in the study felt that their screening rates would improve with the change from Pap smear to CST due to the availability of self-screening and the longer screening interval.

## 4. Discussion

This paper explores health care provider perspectives and approaches to cancer screening of Aboriginal Australians in the NT. There was considerable variation in responses to questions about screening from staff in remote and very remote PHC clinics, sometimes from staff within the same clinic. It is apparent that cancer screening is not consistently promoted or delivered across PHC clinics in the NT, with some staff seeing cancer screening as a “huge gap” due to lack of funding and resources, and others not regarding it as a priority compared to other conditions. Staff at three PHC clinics felt unable to undertake screening due to lack of resourcing and the overwhelming burden of acute and chronic disease, staff at two other clinics reported screening opportunistically during annual health checks, while multiple staff at two other clinics felt they were doing well, with systematic screening, targeted programs and high screening rates. There was large variation in PHC staff perceptions of the breast screening and cervical screening programs, however PHC staff were generally positive about the BreastScreen NT bus and about their clinic’s cervical screening rates. Participants were united in reporting that the bowel screening kit was not appropriate for their Aboriginal patients. The views of the PHC providers aligned with the perceptions of screening by hospital staff. However, there was minimal data about screening from regional hospital and cancer centre staff, possibly reflecting that prevention and early detection are not considered a core part of the role of tertiary health service staff [33].

Our research suggests that cancer screening may not be easily accessible for Aboriginal people living in remote and very remote areas of the NT. Several participants were concerned with the lack of access to screening and diagnostic services (such as mammograms and medical imaging), particularly in remote and very remote communities. Australians living in remote areas generally have poorer access to health care than those living in major cities [34], and the NT has poorer access to health services and cancer services compared to other jurisdictions in Australia [35]. Improved access to cancer screening is needed to improve cancer outcomes, particularly for remote, socially disadvantaged, or vulnerable populations [27,36,37], and this is a focus area in the recently published Australian Cancer Plan [38] and the Aboriginal and Torres Strait Islander Cancer Plan [39]. While travel for screening is not covered by PATS in the NT [40] and is not generally eligible for assistance in other Australian jurisdictions [41,42], staff suggested funding travel for cancer screening in the NT would remove one barrier to accessing screening. This has been implemented in South Australia, where BreastScreenSA is an approved medical specialist service eligible for PATS subsidies [43]. This support aligns with a recommendation in the Aboriginal and Torres Strait Islander Cancer Plan which states that “in circumstances where screening infrastructure is not available, it is important that Patient Assistance Travel Schemes include cancer screening in their programs” [39]. Other strategies that may improve the accessibility of screening include funding PHC clinics to provide transport to screening services, increasing the number of screening services in regional and remote “hub” towns and increasing the frequency of visits to remote and very remote communities by mobile screening programs [27,44]. Another strategy could be more holistic, removing the siloed approach in national screening programs as recommended by the Aboriginal and Torres Strait Islander Cancer Plan, which states that screening programs should be offered in one session if possible [39]. A Canadian study reported increased cervical and bowel cancer screening participation by women living in remote First Nations, Metis, and Hutterite communities through including cervical and bowel cancer screening into the existing mobile breast cancer screening program [45]. Cancer screening could also be offered to patients who have travelled to Darwin for a medical specialist appointment, a strategy temporarily adopted in the NT when the BreastScreen NT Bus program was on hold during the COVID-19 outbreak and which was considered successful by staff associated with the program but was discontinued when the bus service resumed [46].

Several PHC staff in this study did not see cancer screening as a priority compared to other conditions and their high workload. Cancer screening was not a focus in the CARPA Standard Treatment Manual at the time the interviews were conducted, however cervical and breast cancer screening were covered in detail in the Minymaku Kutju Tjukurpa Women’s Business Manual [47,48,49]. It should be noted that breast and bowel cancer screening programs are not executed within the PHC clinic, so educating, promoting, and supporting attendance at screening are time consuming activities with little evident outcome for PHC staff already in “survival mode”. For staff to have capacity to engage patients with screening, primary health care in the NT needs to be adequately resourced for the challenges of geography, provision of culturally safe care, and the burden of disease within the NT Aboriginal populations. Rural and remote health is critically underfunded in Australia, with a recent report finding an annual rural health spending deficit of $6.5 billion [50,51], and remote PHC in the NT is substantially underfunded by Australia’s universal health insurance scheme [52,53]. Needs-based PHC funding is required to improve population health [53]. Furthermore, it is well documented that NT has struggled with recruiting and retaining health professionals, and has increasingly relied on short-term, agency or locum staff, particularly in remote areas [54,55,56].

The high turnover in clinics and reliance on short-term, agency or locum staff, was seen as negatively impacting Aboriginal patients’ trust in clinic staff and their willingness to participate in screening. This is supported by another NT study which found that high turnover negatively impacted cultural safety and continuity of care and led to poorer health outcomes for NTs Aboriginal peoples [57]. While increased funding can help with workforce issues in the short term, supplementary strategies such as such as training to work in a remote setting, additional management and clinical support, opportunities for professional development, and programs to help health professionals build social connections in remote communities, are required for long-term improvements to recruitment and retention [58,59]. It is also vital to increase the number of Aboriginal health professionals working in PHC and supporting cancer screening in the NT. Aboriginal staff are critical to improving the cultural safety of health care and screening, and existing Aboriginal health staff need to be in a supportive and culturally safe workplace [35]. Again, the Australian Cancer Plan and the Aboriginal and Torres Strait Islander Cancer Plan both emphasize the importance of recruiting, training, and retaining a strong Aboriginal workforce across both primary healthcare and cancer care [38,39]. Additional education for health professionals on the incidence of cancer in the Aboriginal population and the importance of cancer screening may be beneficial, particularly as the NT workforce overall is younger, less experienced, and more likely to be employed on short-term, agency or locum contracts compared to other jurisdictions in Australia [54,55,56]. Education on culturally safe ways to talk to patients such as occurs in clinical yarning [60] or medical yarn ups [61] and ways to apply these models to cancer screening could be of benefit, as a number of participants reported struggling to engage their patients on this topic. Clarification may also be needed on the term “screening” as this term can be synonymous with STI screening for many health professionals and members of the community, which can cause confusion when talking about cancer screening [62].

One way for PHC staff to increase cancer screening with their patient cohort would be to routinely recommend and if possible, undertake screening during annual health checks for their Aboriginal patients, as was described by some participants, and which is now included in the Adult Health Check checklist of the *CARPA Standard Treatment Manual for remote and rural practice* 8th edition [47]. All Aboriginal and Torres Strait Islander people are eligible for a free, annual Indigenous-specific health check (also known as the 715 health check) [63] and the recently published Aboriginal and Torres Strait Islander Cancer Plan calls for healthcare teams to be adequately funded and resourced to include cancer screening components in the annual health check [39]. This needs to be supported by good templates within the electronic Patient Information Systems record of primary care. Health check templates for adults aged under 50 years do not prompt questions on breast cancer screening or bowel cancer screening, due to the national screening programs starting at 50 years of age [47,64]. However, Aboriginal Australians are at greater risk of developing these cancers at a younger age than the general population and a growing body of research is calling to lower the bowel and breast cancer screening start age to 40 years for this group [65,66,67,68]. Incorporating more questions on cancer screening into the templates for adults (under 50 years of age) might serve as a prompt for PHC staff to identify high risk patients and start conversations about cancer screening at a younger age, including promoting screening to patients over 40 years of age, as well as raising awareness of the incidence of cancer with this population group.

Providers’ reports that the NBCSP is complicated and not culturally appropriate reiterates findings from multiple studies over the past two decades and repeated calls to improve the cultural appropriateness of the program [68,69,70]. In response, the Australian government has recently permitted PHC providers to bulk order and issue bowel screening kits directly to eligible patients because “patients are more likely to do the test when it has been discussed with a trusted health professional who can explain the test and why it is important” [71]. However, one clinic in this study was so under-resourced and swamped with acute presentations that the box with bowel screen kits had not even been opened. Reports from multiple other clinics in this study, as well as from PHC staff in a previous study [72] and in the National Indigenous Bowel Screening Pilot [73], suggests that additional resourcing, management support, and training are required to support PHC staff to engage with this model. At the time of this data collection, formal communications with GPs and Aboriginal Health Workers has been limited, despite their pivotal role in supporting the program and encouraging uptake amongst Aboriginal Australians [16]. Targeted communications to increase GP and health worker awareness of their roles in the screening program are needed, as are improved linkages with PHC. In addition, locally developed and co-designed community education programs and resources are required to increase knowledge of bowel cancer and confidence to carry out the self-screening test, as well as to help address the shame and stigma associated with bowel cancer screening in the Aboriginal community [16,68,74].

The mobile breast cancer screening program was seen to raise awareness and resulted in earlier diagnosis of breast cancer through two-yearly visits to communities, a finding also reported by a Western Australian (WA) study [26]. Issues with the bus were predominantly around the location being inappropriate in some cases, the lack of community engagement, or lack of access. Consulting with community leaders as well as PHC staff about the most culturally appropriate location of the screening bus is vital to maximize uptake [46]. Currently, BreastScreen NT has one bus servicing two remote towns annually and fifteen remote communities every two years; this contrasts with WA, which has two permanent regional screening clinics and four mobile vans which visit 100 rural locations every two years (with some towns visited annually) [26,75]. Given the size and dispersed population of the NT, the NT is underserviced and may benefit from the establishment of permanent screening clinics in remote “hub” towns such as Katherine and Nhulunbuy (Gove), which would then free up the bus to visit additional remote communities. Adding a second bus to the program would increase the program’s reach and frequency although the fragile nature of breast screening equipment and many unsealed roads in the NT limit the number of very remote communities a mobile screening service could access [44]. Two studies describe how this barrier can be addressed through a collaborative approach which starts with culturally relevant community information sessions on breast health to build awareness and enthusiasm, followed by a group trip to the screening location supported and attended by PHC staff [44,76]. Unfortunately, efforts to increase the number or availability of breast screening services in the NT will be impacted by the national shortage of radiographers trained to provide mammography; a Graduate Diploma qualification that requires applicants to hold a Bachelor degree (or equivalent) in health or biomedical sciences [77,78].

Enablers of cervical screening included targeted programs delivered by nursing staff, proactively contacting eligible women, offering transport to attend the clinic, and education about why cervical screening was important. These strategies are supported by the literature [25,79], with one large Australian cervical screening study also emphasizing the importance of building trusting relationships, employing Aboriginal staff, ensuring female staff are available to conduct cervical screening, and confirming women’s preference for who conducts the screening [25]. In contrast, some providers perceived that long-term relationships with the women in the community sometimes affected the women’s willingness to screen due to feelings of embarrassment caused by the intimate nature of the procedure, a finding also reported by Jaenke et al. [79]. One solution reported by participants was notifying the women when a visiting nurse was available to perform their screening. More recently, the expansion of self-collection eligibility in 2022, which made self-collection available for all people with a cervix, may assist such patients, with recent Australian studies finding self-collection is acceptable to Aboriginal women and “avoided the shame, embarrassment and discomfort surrounding the clinician-collected sample” [80,81]. Improving cervical screening rates and increasing access to screening are key components of Australia’s commitment to achieve equitable elimination of cervical cancer as a public health problem by 2035, which will potentially make Australia the first nation in the world to achieve elimination [82,83].

A National Lung Cancer Screening Program has recently been announced by the Australian Government for commencement by July 2025 [84]. The program will use low dose computed tomography (LCDT) to detect lung cancer earlier in individuals who do not have symptoms and are at higher risk of the disease, including Aboriginal people aged 50 to 74 years [85]. This program aims to reduce health inequities for Aboriginal Australians [85,86] but will create additional challenges for under resourced PHC clinics that are already struggling to meet existing screening guidelines. It is likely that PHC staff will be required to identify eligible patients, educate them on the new program, and assist them to access screening. Our study reinforces previous findings that Australia’s primary care and cancer care workforces are inadequately prepared to provide patients with personalized cancer risk information and advice [19,87]. Access to LCDT scanners is also likely to be challenging for remote communities, requiring either lengthy travel or a comprehensive mobile screening service. However, lessons may be learned from the US, where mobile lung screening programs have been successful at increasing screening rates for underserved populations such as Native Americans and isolated rural groups [88,89].

Screening for chronic HBV, which occurs at higher rates in older Aboriginal people in the NT and underpins higher rates of HCC, was reported to be not meeting national guidelines by one participant in this study. However, the Partnership Approach to Sustainably eliminating Chronic Hepatitis B in the Northern Territory (Hep B PAST) partnership project commenced in 2018 and since then the cascade of care for those living with hepatitis B in the NT has significantly improved, now exceeding National Hepatitis B Strategy Targets [90]. This project aims to eliminate chronic HBV from the Aboriginal population of the NT through a multi-faceted approach incorporating an audit to determine HPV serology for at least 80% of the Aboriginal population in the NT, education to improve HBV health literacy amongst Aboriginal communities and PHC staff, and by transitioning chronic HBV care to PHC clinics using the chronic disease model [91]. The mobile “one-stop liver shop” to improve chronic HBV care described by Hla et al. [92] could also be expanded to other conditions, and the importance of PHC in coordinating this approach must be emphasized.

### 4.1. Limitations

This study took place in the NT, an area of Australia with unique population characteristics [93]. The findings may have relevance for other parts of Australia with significant rural and remote Aboriginal populations, and for countries with large and seasonally inaccessible geographic areas and with similar histories of colonisation and marginalisation of the Indigenous population, including Canada, the United State of America (USA), and New Zealand.

While the number and remoteness of health care professionals in this study is a unique contribution, our efforts to obtain the perspectives of Aboriginal patients were not successful. The team member who conducted the interviews made many attempts to arrange interviews with Aboriginal people affected by cancer, with some agreeing to participate but then not being available on the day due to poor health, recent discharge, or other commitments. The interviewer had lived and worked as a clinician in the NT for many years however they are not Indigenous which may have impacted Aboriginal people’s willingness to participate. However, our interviews included 14 Aboriginal staff among the 50 staff from a diverse range of locations and professions. All PHC clinics in this study were operated by NT Health, as we were unsuccessful in our efforts to obtain the perspectives of staff working at ACCHS. Between June 2017 and June 2019, there were 17 ACCHS in the NT and 53 clinics operated by NT Health, with clinics operated by NT Health seeing just under half (47%) of all Aboriginal regular clients in the NT [94].

Although data collection for this study occurred over a discrete period, health services and PHC clinics are dynamic with respect to policies and programs. Furthermore, data analysis and write up were delayed for some time due to changes in the project team. Consequently, we are aware of new initiatives and guidelines that have been implemented since the interviews were conducted which undoubtedly affect how cancer screening is conducted in the NT, such as the publication of the 8th edition of the *CARPA Standard Treatment Manual* in 2022, which clearly includes questions about cancer screening as part of the Adult Health Check checklist [47].

Another limitation is that transcripts were not returned to participants for member checking. The inclusion of such a participant check may have enhanced the trustworthiness of the data. However, the comments and experiences with respect to screening were repeated as analysis proceeded, and this saturation gives confidence to the conclusions reached.

### 4.2. Recommendations

There is much to be learned from heath care providers perspectives on the provision of cancer screening. We have developed a list of opportunities for the Australian Government and the NT Government (Table 2).

## 5. Conclusions

Cancer screening by remote and very remote PHC clinics in the NT is variable, with some clinics seeing it as an area where they are performing well, while others see cancer screening as a “huge gap”, or lower priority compared to other conditions due to lack of funding and resources. PHC providers in the NT are essential to efforts to improve cancer outcomes for the Aboriginal population by improving access to cancer screening, however they are hampered by lack of funding and resources, high workloads and staff turnover. Our findings suggest that system-level improvements are required, including increased funding and resourcing for screening programs to improve their reach and frequency, increased transport to screening services, adequate resourcing for PHC clinics to enable staff to provide screening, and education for PHC staff on the incidence of cancer in the Aboriginal community, the importance of cancer screening and how to engage Aboriginal patients with screening. Additional resources would assist PHC clinics to incorporate a greater emphasis on cancer screening into adult health checks and would support PHC clinics to work with local communities to co-design targeted programs and culturally relevant education. Addressing these issues will go some way towards addressing barriers to screening for Aboriginal people in the NT. Furthermore, it is vital that these issues are addressed prior to the implementation of the National Lung Cancer Screening Program in 2025 if it is to be successful in the NT and to enable the Australian Government to meet its pledge to be the first nation in the world to eliminate cervical cancer as a public health problem by 2035.

## Figures and Tables

**Table 1 ijerph-21-00141-t001:** Characteristics of participating cancer service providers.

Health Service	Remoteness Area	No. of Participants
Aged Care	Very remote	1
Cancer Centre	Outer regional	6
Cancer Support Service	Outer regional	1
Hospital	Outer regional	3
Hospitals	Remote	4
Palliative Care	Outer regional	2
Primary health care clinic	Outer regional	2
Primary health care clinics	Remote	9
Primary health care clinics	Very remote	22
Total		50

**Table 2 ijerph-21-00141-t002:** Authors recommendations based on health care provider perspectives on the provision of screening to Aboriginal Australians in the NT.

Finding	Recommendation *
Cancer screening is not easily accessible for Aboriginal people living in remote and very remote areas of the NT due to lack of access to screening and diagnostic services, and the cost of travelling large distances to access those services.	• Fund and support NT cancer screening services to utilize co-design approaches, that value local consumers and service providers, to reconceptualize how cancer screening programs might best be delivered in the NT within the parameters of the national program. • Adequate resources to implement a co-design approach will be vital during the imminent roll out of the national Lung Cancer Screening Program. • Adopt a more holistic approach to screening by removing the siloed approach in national screening programs. • Formally trial a program whereby cancer screening is systematically offered to patients who have travelled to a regional centre for a medical specialist appointment. • Offer cervical and bowel cancer screening to attendees of the existing mobile breast cancer screening program. • Add screening services (BreastScreen NT) as an approved medical specialist service eligible for NT PATS subsidies • Fund PHC clinics to provide transport and support for community members to travel to screening services to help reduce the barriers to accessing screening.
PHC staff in this study did not see cancer screening as a priority compared to other conditions and their high workload.	• Increase funding, resourcing and staffing of remote and very remote primary health care to achieve parity with non-remote communities in Australia. • Develop an educational intervention for PHC staff to cover incidence of cancer, the importance of cancer screening and how to engage Aboriginal patients with screening, including explaining why screening is beneficial. • Fund PHC clinics to employ a cancer screening coordinator (or remunerate an existing staff member to take on the role) to manage and coordinate programs locally. • Fund NT cancer screening services to support PHC clinics to have a greater focus on health promotion and raising of health awareness in the community, including through sharing successful strategies from other PHC clinics. Encourage PHC clinics to collaboratively design and develop localised strategies with the community to increase participation in screening programs.
The high turnover in clinics and reliance on short-term, agency or locum staff, negatively impacted Aboriginal patients’ trust in clinic staff and their willingness to participate in screening.	• Develop additional strategies to recruit and retain the PHC and screening workforce, including a focus on increasing the number of Aboriginal health professionals working in PHC and cancer screening. Implement a training program for short-term, agency, and locum staff to ensure they are equipped to deliver screening and health promotion before they commence work in remote PHC clinics.
PHC staff can increase cancer screening with their patient cohort by routinely recommend and if possible, undertake screening during annual health checks for their Aboriginal patients.	• Support and adequately resource remote and very remote PHC clinics to include education and incorporate cancer screening programs into adult health checks, including promoting/offering bowel and cervical cancer screening.
The mobile breast cancer screening program was seen to raise awareness and resulted in earlier diagnosis of breast cancer; however lack of access was an issue.	• Establish permanent breast cancer screening clinics in additional remote hub towns enabling the existing mobile screening service to visit additional remote communities. • Provide incentives and support to encourage health professionals to undertake mammography training to counteract the critical shortage of mammographers in the NT.

* Recommendations were developed by the authors based on the outcomes of the qualitative analysis and were confirmed with key stakeholders involved with cancer screening in the NT and cancer screening in the Aboriginal population.

## Data Availability

The datasets generated and/or analysed during the current study are not publicly available due to small participant numbers and protection of confidentiality. Aggregate data is available from the corresponding author on reasonable request.

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
