# Peer review of "“A Huge Gap”: Health Care Provider Perspectives on Cancer Screening for Aboriginal and Torres Strait Islander People in the Northern Territory"

_ijerph, 2024, doi:10.3390/ijerph21020141_

Round 1

Reviewer 1 Report

Comments and Suggestions for Authors

The manuscript presents a thorough evaluation of health care provider perspectives on cancer screening for the intended population. While it is very comprehensive, it could be improved with the following:

Page 2, line 70. What is the screening rate of bowel cancer in Aboriginal people?

Page 6, line 238, “got” should be get perhaps.

The data analysis could be improved with the following:

Describe the theoretical perspective of the analytical approach. For example, grounded theory, etc. This would aide in the assessment of the data.

How were roles performed and differentiated? Was coding assessed for interrater reliability for example? How many individuals reviewed the transcripts?

How did the authors assess saturation?

A description of the positionality of the coders would also help. For example, were any of the authors Aboriginal people? How could the identity of the coders impact the findings?

Why were individuals from the communities not included in the data collection rather than just healthcare providers? (understanding that some providers were also Aboriginal people). This seems like a major confounder to understand the lived experience of the intended population.

The discussion could be condensed with less repetition of the results that are well described.

Author Response

Thank you very much for taking the time to review this manuscript. Please find the detailed responses below and the corresponding revisions in tracked changes in the re-submitted files.

Reviewers’ comments

Responses to comments

Reviewer 1:

The manuscript presents a thorough evaluation of health care provider perspectives on cancer screening for the intended population.

We thank the reviewer for these comments.

Page 2, line 70. What is the screening rate of bowel cancer in Aboriginal people?

Unfortunately, accurate data on Indigenous participation in the NBCSP is not available, so must be estimated. The national participation rate for Indigenous Australians is estimated at between 19.5% and 35%.

Available data on Indigenous participation in NBCSP screening in the NT is based on population-based participation rather than the program-based participation used nationally, which makes it hard to compare. Population based participation in the NT has been estimated for Aboriginal people at between 2.0%-2.5% per 100 population vs 11.6%-11.8% for non-Aboriginal people. These statistics would cause confusion as they are much lower than the program-based participation rates.

Page 6, line 238, “got” should be get perhaps.

Corrected. Thank you for picking up this typo.

Describe the theoretical perspective of the analytical approach. For example, grounded theory, etc. This would aide in the assessment of the data.

We did not impose any preconceived theory or conceptual model prior to data collection. The interpretation and analysis of findings were guided by social constructionist perspectives to help us understand how people interpret and make sense of experiences within the socio-cultural, political and historical context within which they provide care.

How were roles performed and differentiated? Was coding assessed for interrater reliability for example? How many individuals reviewed the transcripts?

We have indicated which team members were involved during the data analysis stage. For example, on line 213:

“Transcripts were reviewed by three team members. Analysis was iteratively conducted through line-by-line review of the transcripts by one researcher (EVT) to identify themes.  Discussions and additional analysis between SCT and EVT refined themes and triangulated staff interviews.”

How did the authors assess saturation?

This was addressed in the Limitations section, line 754, where we state:

“However, the comments and experiences with respect to screening were repeated as analysis proceeded, and this saturation gives confidence to the conclusions reached.”

A description of the positionality of the coders would also help. For example, were any of the authors Aboriginal people? How could the identity of the coders impact the findings?

One of the authors is a Kamilaroi woman, a professor of Indigenous Health Research and a leader in Indigenous cancer research in Australia.

We have added a description of the positionality of all authors starting line 160:

“The research team involved in the analysis consisted of five women (ET, SD, CC, GG, SCT). GG is a Kamilaroi woman and professor of Indigenous Health Research, with extensive research expertise in improving outcomes for Aboriginal people with cancer. Three of the non-Indigenous team members have clinical backgrounds (SD, CC and SCT). SD and CC are based in the NT and have extensive experience in the provision of Aboriginal health care, primary health care and expertise in cancer screening. SCT and ET have combined experience with collaborative research into improving Indigenous health outcomes of over thirty years.” 

This article has been reviewed by eight staff working in a variety of roles relating to cancer screening in the NT, including three Aboriginal staff. They confirmed that the issues identified and recommendations made by this article are still relevant.

Why were individuals from the communities not included in the data collection rather than just healthcare providers? (understanding that some providers were also Aboriginal people). This seems like a major confounder to understand the lived experience of the intended population.

The team member who conducted the interviews had lived and worked in the NT for many years as a remote area nurse and she was persistent in efforts to interview Aboriginal community members. Unfortunately, she was unsuccessful in this, despite having an Advisory Group for the research which included Aboriginal people and staff employed within ACCHS.

We have provided additional detail on this in the Limitations section of the paper from line 730:

“While the number and remoteness of health care professionals in this study is a unique contribution, our efforts to obtain the perspectives of Aboriginal patients were not successful. The team member who conducted the interviews made many attempts to arrange interviews with Aboriginal people affected by cancer, with some agreeing to participate but then not being available on the day due to poor health, recent discharge, or other commitments. The interviewer had lived and worked as a clinician in the NT for many years however they are not Indigenous which may have impacted Aboriginal people’s willingness to participate.”

There have been challenges for researchers in other studies to capture the perspectives of Aboriginal people with cancer.

However, the aim of this article was to analyse health care provider perspectives and approaches to the provision of screening to identify insights into screening provision and health provider attitudes. This information is important since these are the people who interface with Aboriginal community members and have a role in implementing policy into practice.

The discussion could be condensed with less repetition of the results that are well described.

We have gone through the discussion and removed repetition of results.

Reviewer 2 Report

Comments and Suggestions for Authors

This article is part of a project conducted with government primary healthcare providers in Australia's Northern Territory between 2016 and 2019 around cancer treatment. It reports on information provided by interviewees about cancer screening, which emerged as an area of cancer care that was not considered in the primary research questions. Themes around screening show significant variation between interviewees. The article's discussion and conclusion address how cancer screening could be increased, both to meet standard guidelines and to meet perceived additional screening aspirations based on perceived excess risk among Aboriginal people.

Abstract

Thorough description of the work. However the abstract omits that the interviews used for this project were about cancer programs and treatment, and not specifically about screening.

Introduction

Comprehensive and well written, describing cancer in Aboriginal people in Australia’s Northern Territory, and cancer screening in Australia. However the implementation date of each of Australia's cancer screening programs is not discussed, and it is possible that the well-established cervical cancer screening program of many decades is better integrated into services than the more recent bowel cancer screening program and this may explain the contrasting findings around each of these programs.

Methods

This project involved extraction of data about cancer screening from interviews about cancer treatment from health practitioners whose role is cancer treatment. There was no recruitment of key informants involved in cancer screening.  Can the authors be confident that they have included the views of staff with detailed understanding of screening goals, processes, outcomes and monitoring? 

None of the interview questions mention cancer screening. Can the authors be confident that they have been informed about screening in sufficient detail to write a report solely about screening?

Overall the methods as described do not seem adequate for the project aims, to explore cancer screening. Extraction of data that was not in the original research concept risks bias, analogous to data dredging in statistical analysis.

Results

Selection of interviewees based on involvement in the cancer journey after diagnosis, and lack of questions specifically about screening needs to be addressed in relation to the results.

For example “Staff observed that funding, programs and support were more readily available once a cancer diagnosis was made”. This may indicate that interviewees were more aware of funding for treatment, because they were the target group for the research, rather than practitioners whose role was dedicated to screening.

Although this is qualitative work it may be useful to explore and compare data by remoteness, profession, age, time in role and Aboriginality. Particularly with the discussion about high staff turnover, time in role seems an important area to highlight.

Discussion

The article presented to me contains a comment that the discussion contains repetition of the results. However discussion I have received has almost no reference to the results but leads directly into strategies to improve screening coverage.

The discussion presents a series of evidence-informed recommendations to increase cancer screening among Aboriginal people in the Northern Territory. Although this relates to the research findings about cancer screening, interviewees were not selected for their involvement in cancer screening and not asked about screening. Thus the recommendations are based primarily on other literature rather than the research findings.

There is no analysis of the results, such as comparison with other work, particularly in First Nations populations elsewhere, or other populations in Australia. Triangulation of qualitative data is needed to verify that perceived low screening coverage is consistent with data on coverage because clinicians may not recognise how their role fits into the bigger picture. This is especially true in this work which did not select staff involved in screening, nor ask questions about screening.

Also missing was discussion about screening in the overall cancer journey. How is the cancer journey different for patients diagnosed through screening from those who presented with symptoms?

Also, while it is fair to report health professionals’ beliefs about attitudes of Aboriginal community members, the voices of community members themselves are missing. Assumptions about their beliefs may be incorrect or limited. I note in the section on limitations that community members with cancer declined to participate in research on cancer treatment. However involvement in cancer screening is much more widespread than the experience of cancer care, and it may have been possible to recruit community members who had participated in screening. Thus the experience of cancer screening is missing from this article and needs to be explored alongside descriptions of how healthcare practitioners perceive their patients would experience screening.

The shame of bowel cancer screening, compared with a degree of acceptance and normalisation of cervical cancer screening may reflect the greater maturity of cervical cancer screening (implemented 1991) compared to bowel cancer (implemented 2006). This is nearly an entire generation difference, and it now over 30 years since cervical cancer screening was implemented, and 14 years since bowel cancer screening began.

I wonder whether this discussion would be more appropriately revised as an independent literature review on increasing participation in cancer screening rather than attempting to link it to interviews that were primarily about cancer treatment. 

Limitations

The local data collection in NT is a feature rather than a limitation of the research, and the focus is on implementation rather than generalisation.

Delay in publication from the dates of data collection 2016 to 2019 reduces the importance of the work because of significant changes in cancer screening introduced across Australia in the interim. Interviewees

The key limitations in my view are the selection of interviewees for their role in cancer treatment while this article is about cancer screening, together with the aim to explore interviewees’ understanding of cancer screening despite the lack of questions about cancer screening in the interviews.

The fact that all healthcare services involved were government rather than Aboriginal community controlled primary health care organisations (ACCHOs) is also a limitation. It would be useful to have an estimate of the proportion of the Aboriginal population who attend government services, ACCHOs, those who attend both and those who attend neither so we can appreciate the extent of this limitation, particularly in relation to cancer screening.

The absence of Aboriginal community members as participants is a key weakness. This reflects that this work is secondary analysis of data on cancer treatment from treatment providers, rather than community perceptions of the screening programs.  

Recommendations and conclusion are strong but these are based on literature review rather than the qualitative data of this research.

References

References were affected by the delay from data collection to submission, I note reference 20 references to an internet access in 2021, which can no longer be verified.

Reference 59 is incomplete.

Please review the personal communication as a reference (45), as it is possible that this is now published.

Author Response

Thank you very much for taking the time to review this manuscript. Please find the detailed responses below and the corresponding revisions in tracked changes in the re-submitted files.

Reviewers’ comments

Responses to comments

Reviewer 2:

Abstract

Thorough description of the work. However the abstract omits that the interviews used for this project were about cancer programs and treatment, and not specifically about screening.

We apologise that the way we worded the paper led the reviewer to believe that the interviews contained no questions on screening. The interviews collected data about the whole cancer pathway, including prevention and early detection. We have reworded the abstract starting line 17 to clarify that the study was not focused solely on screening:

“This paper analyses health care provider perspectives and approaches to the provision of cancer screening to Aboriginal people in the NT that were collected as part of a larger study.”

Introduction

Comprehensive and well written, describing cancer in Aboriginal people in Australia’s Northern Territory, and cancer screening in Australia.

We thank the reviewer for these comments.

However the implementation date of each of Australia's cancer screening programs is not discussed, and it is possible that the well-established cervical cancer screening program of many decades is better integrated into services than the more recent bowel cancer screening program and this may explain the contrasting findings around each of these programs.

We have added the implementation date of each of Australia’s cancer screening programs on line 60: 

“Australia has national population-based cancer screening programs for breast, cervical and bowel cancer [9]. The National Cervical Screening Program (NCSP) began in 1991; the BreastScreen Australia program was introduced between 1991 and 1995; and the National Bowel Cancer Screening Program (NBCSP) started in 2006 but was not fully implemented until 2020 [10].”

We have expanded on the delays in rolling out the NBCSP on line 89:

“When the NBCSP began, eligible participants in two age bands, 55 and 65 years, were offered a one-off test and in 2008, a one-off test for people turning 50 was added to the program; from 2013 the program was slowly expanded with biennial screening of all Australians aged 50-74 by 2020 [16].”

We have added a brief discussion of the possible impact of the implementation date on line 106:

“It should be acknowledged that the NBCSP was launched 15 years after the other national population-based screening programs and had an incremental roll out. This delayed comprehensive education campaigns, directed at the public and at health professionals. It is also the only self-administered population screening test and the first population screening test to include men, which may contribute to the lower participation rate [10].”

Methods

This project involved extraction of data about cancer screening from interviews about cancer treatment from health practitioners whose role is cancer treatment. There was no recruitment of key informants involved in cancer screening.  Can the authors be confident that they have included the views of staff with detailed understanding of screening goals, processes, outcomes and monitoring?

Again, we regret that our wording in the paper misled the reviewer to believe that we only interviewed health practitioners involved in cancer treatment. We have reworded sections that gave this impression, including related to data collection. As shown in Table 1, 33 of the 50 participants (66%) worked in primary health care, and not cancer treatment services. Hence, they had a role in promoting cancer screening and detection, and in managing patients post treatment.

The sampling section is described in the study methods as “Relevant service providers engaged in providing or overseeing health and supportive care of Aboriginal patients and their families”.

We have clarified our selection and recruitment process:

Line 170:

“Health professionals and support staff (Aboriginal or non-Aboriginal) in the NT were eligible if they were involved with the treatment, care or support of Aboriginal cancer patients, if they filled a leadership role in the care of Aboriginal cancer patients, or if they provided primary health care to Aboriginal patients and their families.”

The research team included individuals with experience with cancer screening in the NT and they engaged and sought feedback extensively with colleagues who filled key positions in the provision of cancer screening services in the NT.  Many of these individuals are explicitly named in the Acknowledgements section of the paper. We have added the following from line 230:

“Data interpretation and recommendations arising from the data were checked with key stakeholders involved with cancer screening in the NT and cancer screening in the Aboriginal population and additional information incorporated into the final analysis and recommendations.”

None of the interview questions mention cancer screening. Can the authors be confident that they have been informed about screening in sufficient detail to write a report solely about screening?

While the interview guide contains no specific questions on screening, it was a guide for “general areas to be discussed”. The interviewer explored the entire cancer pathway, including prevention, early detection, and end-of-life care and prompts were modified to suit the context in which the person being interviewed worked. Questions about screening were raised by the interviewer for interviewees working in primary care if the issue was not first raised by the health professional.

We have clarified this in the data collection section line 196:

“The interviews were guided by a set of broad, open-ended enquiries including questions on the staff member’s role in providing care to Aboriginal people, the typical pathway for Aboriginal cancer patients, how decisions about cancer treatment are made, links between primary and tertiary health services, and questions around primary prevention, early detection and screening, community education and end-of-life- care.”   

Overall the methods as described do not seem adequate for the project aims, to explore cancer screening. Extraction of data that was not in the original research concept risks bias, analogous to data dredging in statistical analysis.

Our explanations above address this issue.

Results

Selection of interviewees based on involvement in the cancer journey after diagnosis, and lack of questions specifically about screening needs to be addressed in relation to the results.

For example “Staff observed that funding, programs and support were more readily available once a cancer diagnosis was made”.

This may indicate that interviewees were more aware of funding for treatment, because they were the target group for the research, rather than practitioners whose role was dedicated to screening.

As previously explained, interviewees were selected from services involved across the cancer journey. The majority of participants (66%) were from primary health care, and were involved in cancer screening and detection, and managing patients post treatment.  We have clarified this in the article.

Questions specifically about screening were raised by the interviewer. We have clarified this in the article.

Although this is qualitative work it may be useful to explore and compare data by remoteness, profession, age, time in role and Aboriginality. Particularly with the discussion about high staff turnover, time in role seems an important area to highlight.

Where possible we have tried to report the results by remoteness, profession, and Aboriginality. This information about participants is also provided in brackets after quotes used throughout the text. 

As noted by the reviewer, time in role is potentially relevant to the discussion on turnover, so we have added some extra information on this from line 321:

“Of those interviewed, 12% (n=6) had been in their current position for less than 1 year and 22% of participants working in very remote locations had been in their current position for less than 1 year.”

Discussion

The article presented to me contains a comment that the discussion contains repetition of the results. However discussion I have received has almost no reference to the results but leads directly into strategies to improve screening coverage.

I wonder whether this discussion would be more appropriately revised as an independent literature review on increasing participation in cancer screening rather than attempting to link it to interviews that were primarily about cancer treatment.

We respectfully beg to differ with the reviewer. We feel that every paragraph of the discussion refers to the results and compares those results with other publications.

We are supported in this by the Editor’s comment that our discussion section contains “a lot of repeating of the results” and the comment by Reviewer 1 that the “discussion could be condensed with less repetition of results that are well described”.

The discussion presents a series of evidence-informed recommendations to increase cancer screening among Aboriginal people in the Northern Territory. Although this relates to the research findings about cancer screening, interviewees were not selected for their involvement in cancer screening and not asked about screening. Thus the recommendations are based primarily on other literature rather than the research findings.

As previously explained, interviewees were selected from across the cancer journey and were asked about screening. 

To clarify how recommendations are based on the research findings, we have presented our recommendations in a table with clear links to the findings.

There is no analysis of the results, such as comparison with other work, particularly in First Nations populations elsewhere, or other populations in Australia.

We respectfully disagree with this comment of the reviewer. We feel that every paragraph of the discussion analysis the results and compares them to other work.

Specific examples in the Discussion include (but are not limited to):

·         “The high turnover in clinics and reliance on short-term, agency or locum staff, was seen as negatively impacting Aboriginal patients’ trust in clinic staff and their willingness to participate in screening. This is supported by another NT study which found that high turnover negatively impacted cultural safety and continuity of care and led to poorer health out-comes for NTs Aboriginal peoples [57].”

·         “Providers’ reports that the NBCSP is complicated and not culturally appropriate, reiterates findings from multiple studies over the past two decades and repeated calls to improve the cultural appropriateness of the program [69-71].”

·         “The mobile breast cancer screening program was seen to raise awareness and resulted in earlier diagnosis of breast cancer through two-yearly visits to communities, a finding also reported by a Western Australian (WA) study [26].”

·         “Enablers of cervical screening included targeted programs delivered by nursing staff, proactively contacting eligible women, offering transport to attend the clinic, and education about why cervical screening was important. These strategies are supported by the literature [25,80]…”

Triangulation of qualitative data is needed to verify that perceived low screening coverage is consistent with data on coverage because clinicians may not recognise how their role fits into the bigger picture.

There is already considerable information on the low screening coverage in the NT in the introduction.

Also missing was discussion about screening in the overall cancer journey. How is the cancer journey different for patients diagnosed through screening from those who presented with symptoms?

That is not the topic of this paper or a question that can be addressed with our data.

Also, while it is fair to report health professionals’ beliefs about attitudes of Aboriginal community members, the voices of community members themselves are missing. Assumptions about their beliefs may be incorrect or limited. I note in the section on limitations that community members with cancer declined to participate in research on cancer treatment. However involvement in cancer screening is much more widespread than the experience of cancer care, and it may have been possible to recruit community members who had participated in screening. Thus the experience of cancer screening is missing from this article and needs to be explored alongside descriptions of how healthcare practitioners perceive their patients would experience screening.

The team member who conducted the interviews had lived and worked in the NT for many years as a remote area nurse and she was persistent in efforts to interview Aboriginal community members. Unfortunately, she was unsuccessful in this, despite having an Advisory Group for the research which included Aboriginal people and staff employed within ACCHS. We have provided additional detail on this in the Limitations section of the paper from line 730:

“While the number and remoteness of health care professionals in this study is a unique contribution, our efforts to obtain the perspectives of Aboriginal patients were not successful. The team member who conducted the interviews made many attempts to arrange interviews with Aboriginal people affected by cancer, with some agreeing to participate but then not being available on the day due to poor health, recent discharge, or other commitments. The interviewer had lived and worked as a clinician in the NT for many years however they are not Indigenous which may have impacted Aboriginal people’s willingness to participate”

While we acknowledge that the experience of cancer screening is missing from this article, the aim of the article was to analyse health care provider perspectives and approaches to the provision of screening, to identify insights into screening provision and health provider attitudes.

The shame of bowel cancer screening, compared with a degree of acceptance and normalisation of cervical cancer screening may reflect the greater maturity of cervical cancer screening (implemented 1991) compared to bowel cancer (implemented 2006). This is nearly an entire generation difference, and it now over 30 years since cervical cancer screening was implemented, and 14 years since bowel cancer screening began.

We have added a brief discussion on the possible impact of the implementation date to the Introduction, line 106.

Limitations

The local data collection in NT is a feature rather than a limitation of the research, and the focus is on implementation rather than generalisation.

We agree with the reviewer’s point. We have reworded the opening sentence to reflect that this is not a limitation, while also highlighting that the findings may be relevant for other parts of Australia with significant rural and remote Aboriginal populations, and for countries with large and seasonally inaccessible geographic areas and with similar histories of colonisation and marginalisation of the Indigenous population.

Delay in publication from the dates of data collection 2016 to 2019 reduces the importance of the work because of significant changes in cancer screening introduced across Australia in the interim.

We acknowledge that the delay between data collection and publication in this study is not ideal. Reasons for delays in this study were multifactorial and included:

·         Time taken to get to know, work with and gain the trust of the health service personnel, particularly staff working in remote and very remote primary healthcare clinics.

·         Turnover of research team members

·         Covid-19, as staff were working in states affected by lockdown for all of 2020 and 2021. Attempts to re-engage NT staff were initially delayed by the Covid outbreak in the NT in late 2021 and early 2022.

These reasons were mostly outside of our control and could not easily have been averted.

However, we wanted to honour the contributions made by the 50 health personnel who participated in this study. Furthermore, the findings and recommendations of this article have been reviewed by staff currently working in cancer screening in the NT (and co-authors of this article). They confirmed that the issues identified are still very relevant and the recommendations made in this article are important for policy understanding and refinement. Information such as this can be used to highlight issues and advocate for changes in resourcing and focus.

Interviewees

The key limitations in my view are the selection of interviewees for their role in cancer treatment while this article is about cancer screening, together with the aim to explore interviewees’ understanding of cancer screening despite the lack of questions about cancer screening in the interviews.

As previously explained, interviewees were selected from across the cancer pathway and were asked about screening.  Our apologies for not making this clear in the previous version, we have now clarified this in the article.

The fact that all healthcare services involved were government rather than Aboriginal community controlled primary health care organisations (ACCHOs) is also a limitation. It would be useful to have an estimate of the proportion of the Aboriginal population who attend government services, ACCHOs, those who attend both and those who attend neither so we can appreciate the extent of this limitation, particularly in relation to cancer screening.

It is important to recognise that Aboriginal people are highly mobile and attend multiple services. We agree it is a limitation that we do not have perspectives of providers who work in Aboriginal Community Controlled Health Services (ACCHS), however, again this was not for lack of effort to engage with them.

We have provided some additional information in the Limitations section on the number of NT government-run clinics vs the number of ACCHS and the proportion of the Aboriginal population who attend government clinics from line 740:

“Between June 2017 and June 2019, there were 17 ACCHS in the NT and 53 clinics operated by NT Health, with clinics operated by NT Health seeing just under half (47%) of all First Nations regular clients in the NT [98].”

The absence of Aboriginal community members as participants is a key weakness. This reflects that this work is secondary analysis of data on cancer treatment from treatment providers, rather than community perceptions of the screening programs. 

As discussed, we have provided additional detail on this in the Limitations section of the paper from line 732.

While we acknowledge that Aboriginal people’s experience of cancer screening is missing from this article, the aim of the article was to analyse health care provider perspectives to provision of screening as improving screening rates is an important component of improving cancer outcomes.

Recommendations and conclusion

Recommendations and conclusion are strong but these are based on literature review rather than the qualitative data of this research.

To clarify how recommendations are based on the research findings, we have presented our recommendations in a table with clear links to the findings.

References

References were affected by the delay from data collection to submission, I note reference 20 references to an internet access in 2021, which can no longer be verified.

Fixed. I was able to access the URL at the reference in question and have updated the access date. I have also verified all other URLs which previously had an access date of 2021.

Reference 59 is incomplete.

Fixed.

Please review the personal communication as a reference (45), as it is possible that this is now published.

This is not published.

Round 2

Reviewer 2 Report

Comments and Suggestions for Authors

Authors have addressed my concerns, very well.

Concluding recommendations are great.

I am frustrated by the claim that WA PATS covers screening mammography as this is not my experience and undermines efforts of WA practitioners to advocate for breast screening. As in NT, in WA there is a mobile screening service and the PATS guideline claims to support screening where this is not available "within appropriate time frame". The existence of a second yearly mobile service enables PATS to refuse all claims. I do not know the situation in SA but would be wary of assuming that the claims on the website are supported by the experience of local services.

Although cancer screening is not covered in CARPA, both cervical and breast cancer screening are covered in detail in the current Women's Business Manual (WBM) (pp 281 and 297). Please confirm they are not in the version current at the time of your study before claiming that they are not mentioned in CARPA, as CARPA this is not the relevant manual for women's health.

Author Response

Reviewer’s comments

Reponses to comments

Reviewer 2:

Authors have addressed my concerns, very well.

Concluding recommendations are great.

We thank the reviewer for their comments.

I am frustrated by the claim that WA PATS covers screening mammography as this is not my experience and undermines efforts of WA practitioners to advocate for breast screening. As in NT, in WA there is a mobile screening service and the PATS guideline claims to support screening where this is not available "within appropriate time frame". The existence of a second yearly mobile service enables PATS to refuse all claims. I do not know the situation in SA but would be wary of assuming that the claims on the website are supported by the experience of local services.

We thank the reviewer for sharing their experience and acknowledge that claims on brochures do not always equate to actions or support. We have removed the claim that WA PATS covers screening mammography, but left SA in because the SA guidelines clearly state that BreastScreen SA is eligible for PATS subsidies.

We have reworded from line 504:

“This has been implemented in South Australia, where BreastScreen SA is an approved medical specialist service eligible for PATS subsidies [43].”

Although cancer screening is not covered in CARPA, both cervical and breast cancer screening are covered in detail in the current Women's Business Manual (WBM) (pp 281 and 297). Please confirm they are not in the version current at the time of your study before claiming that they are not mentioned in CARPA, as CARPA this is not the relevant manual for women's health.

We thank the reviewer for their attention to detail and for raising this point. We have checked the version of the WBM current at the time of the study (the 6th edition) and both cervical and breast cancer screening are covered in detail.

We have left in the comment about CARPA because it is worth noting that cancer screening is now clearly included in the 8th edition, which is a change from the 7th edition. However, we have reworded from line 528 to acknowledge that screening was covered in the WBM:

“Cancer screening was not a focus in the CARPA Standard Treatment Manual at the time the interviews were conducted, however cervical and breast cancer screening were covered in detail in the Minymaku Kutju Tjukurpa Women’s Business Manual [47-49]”